# Training image classifiers using semi-weak label data

## Abstract

This paper introduces a new semi-weak label learning paradigm which provides additional information in comparison to the weak label classification. We define semi-weak label data as data where we know the presence or absence of a given class and additionally we have the information about the exact count of each class as opposed to knowing the label proportions. A three-stage framework is proposed to address the problem of learning from semi-weak labels. It leverages the fact that counting information is naturally non-negative and discrete. Experiments are conducted on generated samples from CIFAR-10 and we compare our model with a fully-supervised setting baseline, a weakly-supervised setting baseline and a learning from proportion(LLP) baseline. Our framework not only outperforms both baseline models for MIL-based weakly supervised setting and learning from proportion setting, but also gives comparable results compared to the fully supervised model. Further, we conduct thorough ablation studies to analyze across datasets and variation with batch size, losses architectural changes, bag size and regularization, thereby demonstrating robustness of our approach.

## 1 Introduction

In a traditional fully supervised machine learning setting, training samples are "strongly" supervised, i.e. every training instance is labeled. In practice, though, strongly labeled data are expensive to collect. An alternate approach is to collect "weak" labels – labels which only indicate the presence or absence of instances of a class in sets of training samples. This form of labelling is particularly useful for data such as images or sounds. For instance, in an image, it is relatively easy to annotate if the image contains instances of (say) dogs, but much harder to tag the bounding box of every dog in the image. The former represents a weak label, while the latter is a strong label. So also, in sound recordings it is much easier to merely annotate if a recording includes gun shots (weak label) than to identify the onset and offset times of each instance of a shot (strong label).

At an abstract level, it is useful to think of data such as images or sound recordings as *bags* of candidate instances (e.g. candidate regions of the image or candidate sections of the recording). Labels are now assigned to bags, rather than instances. A negative label assigned to a bag indicates that the original data (image or recording) did not contain the target class(es), and hence none of the instances in the bag formed from it are positive for any class. On the other hand, a positive label assigned to a bag indicates that some instances in the bag are positive, although it is unknown which or how many. In the real world, it is much easier to collect such bag-level, or weak labels. A number of algorithms have also been proposed to train classifiers with such weak labels Shah et al. (2018); Pappas & Popescu-Belis (2014); Carbonneau et al. (2018); Vanwinckelen et al. (2016). However, there remains a gap between the performance of fully supervised modelsKumar & Raj (2016); Shah et al. (2018) (trained from data where every instance is labelled) and that of the weakly supervised model (trained from data with weak labels). The gap in performance limits the ability to which weak supervision (i.e. supervision with weak labels) can be relied upon.

In this paper, we introduce a middle ground between the two settings of full and weak supervision. In many settings, it is possible to annotate *count* information, which specifies the number of occurrences of individual classes in a bag, for not much extra effort over simple weak labeling that merely tags their presence or absence. For instance, it is often fairly straight-forward to annotate *how many* dogs there are in an image, if the annotation of exact bounding boxes is not required. Similarly, it is

reasonable to expect that it is not significant extra effort to indicate how many (e.g.) gunshots were heard in an audio recording, if it is not required that their exact locations be tagged as well.

We refer to such labels as "semi-weak" labels, and the problem of learning from such data as as learning with semi-weak supervision, or learning from counts. In our paper, we show classifiers trained using semi-weak labels can classify test instances with much greater accuracy than those trained with merely weak labels.

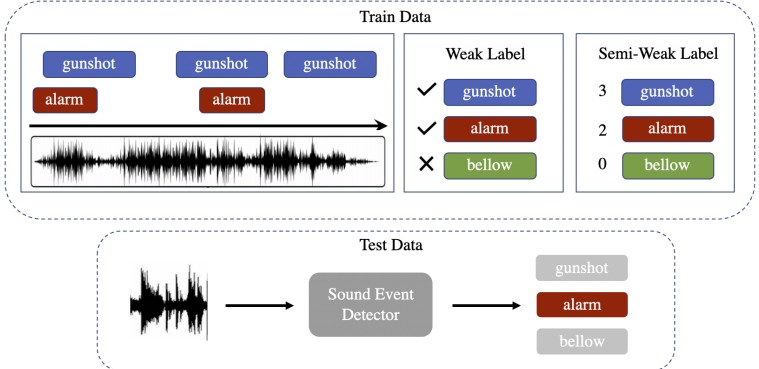

Figure 1: An example of semi-weak labels for audio event detection. Semi-weak labels give count information, while weak labels only provide information about the presence or absence of classes.

Our proposed solution for learning from semi-weak labels is to pose the training as a constraint satisfaction problem. The constraint to satisfy is that the sum of the counts predicted per category in each bag must equal the size of the bag. Thus, we first learn to predict whether the instance is present in the bag or not similar to weak label learning setup and learns an expected count of each class in a bag. Then, given the bag size and conditional expected count for each class, we propose to solve an optimization problem to translate the real valued expected count to non-negative integer count. Finally, given the predicted count as "ground truth" and the instance level logits, we try to find the best assignment of labels such that the counting requirement is satisfied as well as the likelihood of the bag is maximized.

## 2 LITERATURE REVIEW

### 2.1 MULTIPLE INSTANCE LEARNING

Learning from bag-level labels is often referred to as *multiple instance learning* (MIL). Paper Carbonneau et al. (2018) provides an extensive summary detailing most aspects of multiple instance learning. Dietterich Dietterich et al. (1997) first introduced multiple instance learning where it was applied for drug activity prediction. Since then, several algorithms have been introduced to address MIL. To deal with label noise, a natural solution is to count the number of positive instances in a bag, and apply a threshold to tag a bag as positive. This was summarized in Foulds & Frank (2010) as the threshold-based assumption for multiple instance learning. The threshold-based assumption, which defined that a bag is positive if and only the number of positive instances is between a range, was the first time that counting information was brought into multiple instance learning. Since then, several effortsTao et al. (2004b;a) were exerted to do bag-level prediction using count-based assumptions. Foulds & Frank (2010) extended the assumption and proposed an SVM-based algorithm to predict the bag label. One common problem with those methods was scalability; also those methods do not generalize for multi-class classification.

**Multiple Instance Regression**: MIL regression consists of assigning a real value to a bag. Compared to MIL classification, MIL regression has attracted far less attention in the literature. For MIL regression, one line of research has the assumption that some primary instance contributes largely to the bag label. This motivated sparsity-based approaches that assign sparse weights to instances and use regularization methods like L1, L2 regularizers.Pappas & Popescu-Belis (2014); Wagstaff & Lane (2007); Pappas & Popescu-Belis (2017). However, most of these method work only for small-scale data and focus on the accuracy of predicted results rather than attempting to identify the

primary instances that contribute to the bag label. Subramanian et al. (2016) addressed the identification of primary instances by using a Dirichlet process to group the instances and find clusters. However, this method assumes that the largest cluster defines the label; this method also doesn't work for multi-class machine learning problems. Also, those methods do not work properly in our semi-weakly supervised setting as they fail to incorporate the natural property of counts, which is a non-negative integer value.

**Instance-Level Prediction**: Another relevant mainstream of MIL research is instance-level prediction. MaronMaron & Lozano-Pérez (1997) is perhaps the best-known framework for instance-level prediction for MIL. Following this framework, many research ideas have been proposed. The basic ideas of those frameworks are to label the instance dynamically or statically according to the bag label. Training instance-level classifiers is non-trivial as strong labels are unavailable. Recently, many methods have proposed to do bag-level prediction and "hope" that the bag-level accuracy could propagate to instance-level accuracy. Dietterich et al. (1997); Andrews et al. (2003); Babenko et al. (2008). However, as is discussed in Doran & Ray (2014); Vanwinckelen et al. (2016), this method is sub-optimal. Empirical studies have been conducted in Vanwinckelen et al. (2016) that better bag level prediction doesn't promise instance-level prediction. Therefore, the the most successful instance-level prediction models, i.e., mi-SVM and SI-SVMAndrews et al. (2003), discard the bag information as much as possible, and treat each instance individually.

## 2.2 Learning from Proportions

A related approach, previously proposed in the literature, is that of learning from *proportions* Dulac-Arnold et al. (2019); Yu et al. (2013); Musicant et al. (2007), where the proportion of instances in a bag that belong to any class is indicated. When the precise number of instances in individual bags is known, knowing proportions is identical to knowing instance counts of classes in the bag. But in other settings, such as in the image or sound examples mentioned above, it is much simpler, and possibly far more natural to annotate the *counts* of occurrences of a class than the proportions. For instance, in an audio recording which must be labelled for the occurrence of, say, dog barks, it is more intuitive and natural to state that a dog barked three times in the recording than to say (*e.g.*) that 20% of the recording comprised the sound of dogs barking. Indeed, the main signature of sound events such as dog barked is their onset, which is generally distinct, whereas it is generally hard to spot precisely when the sound *terminates*. As such, while it is perfectly reasonable to state that the recording includes three barks, it may be infeasible, or even meaningless to specify what fraction of the recording comprised dog barks as shown in Figure 2.

Quadrianto Quadrianto et al. (2009) proposed the MeanMap model which assumes the data follows an exponential distribution and is conditionally independent of the bags. Fan Fan et al. (2014) and PatriniPatrini et al. (2014) further refined the loss function of MeanMap and make it applicable for multi-class classification. Yu Yu et al. (2014) proposed another line of research and used SVM models to iteratively estimate the instance-level classifier. However, this method suffers from scalability issues when it is extended to a multi-class setting. More recently, a DL-based method was proposed by Dulac-Arnold et al. (2019); Ardehaly & Culotta (2017); Bortsova et al. (2018). The commonality for the DL-based methods for the LLP problem is that they try to use bag-level supervision to perform instance-level estimation such that the estimated distribution is as close as possible to the bag label. Interestingly, Dulac-Arnold et al. (2019) introduces another loss function that directly minimizes the instance-level prediction results. They introduced the Optimal Transport algorithm to make the loss function computationally tractable.

## 3 Problem Statement

Figure 1 shows an example problem setting in audio event detection. Generally, we consider a supervised multi-class classification problem, where we define the instance-level data $x \in \mathcal{X}$ and label $y \in \mathcal{Y}$, where $\mathcal{X} \subset \mathbf{R}^{H \times W \times C}$ and $\mathcal{Y} = \{0, \ldots, K\}$. Let $K$ be the number of classes, $N_B$ be the number of instances in a bag and $N$ be the number of bags. For each bag $b_i \in \mathbf{B} = \{\mathcal{X}\}$, $y_i^B \in \mathbf{Y}^B$ is the bag-level label. More specifically, $y_{i,k}^B$ is the count of class $k$ in bag $i$. Formally,

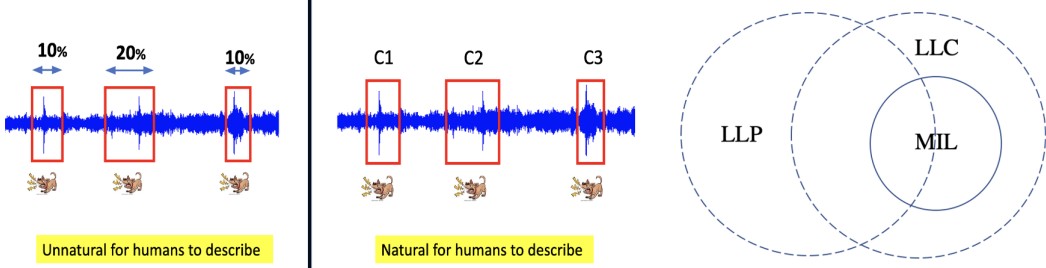

Figure 2: Left: We illustrate scenario which is natural for human to annotate between counts and proportions. Right: The relation between learning from proportion (LLP), learning from count (LLC) and Learning from Weak labels (MIL). The intersection part assumes that the bag size is known, and the count is exact. MIL problem is a special case of LLC where the count is equal to 1.

given the bag size, we define the space of labels as

$$\mathbf{Y}^B = \{c \in \mathbb{Z}_+^K : \sum_{i=1}^K c_i = N_B\} \tag{1}$$

Given a series of bags, $S = [(b_0, y_0^B), (b_1, y_1^B), \dots, (b_N, y_N^B)] \subseteq \mathbf{B} \times \mathbf{Y}^B$, our goal is to learn a predictor $f : \mathcal{X} \to \mathbb{R}^K$ that predicts the probability distribution of an instance belonging to different classes, i.e. $f_\theta(x_i) = \{p(y_i = 0|x_i), p(y_i = 1|x_i), \dots, p(y_i = K|x_i)\}$, where $x_i$ is a feature vector for an instance and $\theta$ is the parameters of the network.

### 3.1 A GENERAL LOSS FUNCTION

In this problem, we focus on instance-level prediction and solve it by viewing the problem as a risk minimization task. More specifically, given a regression loss function $L_{reg} : \mathcal{B} \times \mathbf{Y}^B \to \mathbf{R}_+$ defined as

$$R_N(f) := \frac{1}{N} \sum_{i=1}^N L_{reg}(f(x_i), y_i^B) \tag{2}$$

One intuitive approach is the define a divergence function that quantitatively quantify the distance between the predicted bag labels and the ground-truth label. As such, let's define $\forall b_i \in \mathcal{B}$, the predicted count as $= \sum_{j=1}^{n_i} f(x_j)$, and we want to minimize the divergence between the predicted counts and the actual counts

$$L_{reg}(f(x_i), y_i^B) := \frac{1}{n_i} div(, y_i^B)) \tag{3}$$

Usually, learning from counts could be formulated as a regression problem. However, for inter-class separation of the features at the bag-level and to tackle the problem as multi-label bag classification, we used binary cross-entropy to recognize different classes existing in a bag. So let $1(\cdot)$ denotes the indicator function, taking values 1 or 0 depending on whether its argument is true or not, we define another loss term as follows

$$F_{max} = max_{i \in 0,1,\dots,B_N}[f(x_0), f(x_1), \dots f(x_{B_N})] \tag{4}$$

$$L_{cls}(f(x_i), y_i^B) := div_{bce}(F_{max}, 1(y_i^B > 0)) \tag{5}$$

Additionally, we require the loss function to be robust to the sparsity of the bag and the generalizability of the loss function is critical in the learning process. Thus, we introduce one L1 regularization

term that makes the model perform better when the bag is sparse or "pure". Here, we define sparseness based on the bag level counts of each class. Our proposed loss function with the regularization term is defined as follows

$$R_N(f) = L_{reg} + \alpha L_{cls} + \beta \sum_{x_i \in b_i} \frac{1}{N_B} \|f(x_i)\|_{L1} \tag{6}$$

## 4    PRELIMINARY: DLLP APPROACH

DLLP Ardehaly & Culotta (2017) approach is a DNN-based model for learning from proportion. Mathematically, given a bag $b_1, b_2, \cdot, b_N$, suppose that $f_\theta(b_{ij}) = p_\theta(\mathbf{y} \mid \mathbf{b}_{ij})$ is the vector outputs of the DNN for the $j$-th instance in $b_i$. Let $\bigoplus$ is an element-wise addition operator. Then the posterior bag-level class proportion could be estimated as

$$\overline{\mathbf{p}}_i = \frac{1}{N_B} \bigoplus_{j=1}^{N_B} p_\theta(\mathbf{y} \mid \mathbf{b}_{ij}) \tag{7}$$

The final objective, i.e., $L_{dllp}$, is made of a distance measurement. Let $\mathbf{c}_i^k$ be the counts of class $k$ in bag $i$, $\mathbf{p}_i^k$ be the counts normalized by bag size, and $\overline{\mathbf{p}}_i^k$ be the $k$-th element in the vector $\overline{\mathbf{p}}_i$ (the estimated proportion of class $k$), then the loss is defined as $L_{dllp} = KL(\mathbf{p}_i, \overline{\mathbf{p}}_i)$. $KL$ refers to the commonly used Kullback–Leibler divergence for measuring the distance between two distributions.

## 5    PROPOSED METHODS: TWO-STAGE FRAMEWORK

In this section, we introduce a two-stage model for learning from counts. It is extended from DLLP Ardehaly & Culotta (2017).

### 5.1    STAGE-1: ESTIMATING CLASS COUNT

#### 5.1.1    POISSON LOSS AND EXPECTED COUNT

As mentioned in Section 1, one difference between LLP problem and LLC problem is that the label for LLC problem is discrete. So it is not optimal to use a KL-divergence to measure the difference between prediction and the true labels. Instead, we propose to use Poisson Loss - a Poisson distribution based loss function. The Poisson distribution is the discrete probability distribution of the number of events occurring in a given time period, which applies to the LLC setting if we consider the counting is the number of times a class instance appears in this bag Letkowski (2012).

In our framework, we assume that the counting for class $j$ follows a Poisson distribution, i.e. $p(c_i^j = k|\lambda_j) = \frac{\lambda_j^k e^{-\lambda_j}}{k!}$. For simplicity, since we assume the size of the bag is unknown, we assume the count for each class in a bag is independent. Given a bag of output of the network $p_\theta(\mathbf{y} \mid \mathbf{b}_{ij})$, we define $\overline{\mathbf{c}}_i^k$ as the expected count for class $k$ in bag $b_i$.

$$\overline{\mathbf{c}^k}_i = \bigoplus_{u=1}^{N_B} p_\theta(\mathbf{y} \mid \mathbf{b}_{ij}^u) \tag{8}$$

So we have

$$\hat{\lambda} = \hat{\mathrm{E}}(c_i^k \mid x) = \overline{\mathbf{c}^k}_i \tag{9}$$

Then the Poisson loss is defined as the negative log likelihood function

$$L_{reg}(y, \hat{\lambda}) = -\log(p(y|\hat{\lambda})) = \hat{\lambda} - y\log(\hat{\lambda}) + \log(y). \tag{10}$$

Because the last term in Equation 14 would be a constant for a given bag, it is usually omitted. So the final loss and its gradient is

$$
L(y, \hat{\lambda}) \propto \hat{\lambda} - y \log(\hat{\lambda})
$$
$$
\nabla_{\hat{\lambda}} L(y, \hat{\lambda}) = 1 - y/\hat{\lambda}.
$$

(11)

The following are two interesting properties for the Poisson loss. **Adapted gradient:** Unlike other distance functions like mean absolute error, the Poisson loss doesn't have a constant gradient for all input values. Also, when the actual count is large, the gradient value would be relatively smaller. This meets our intuition that when the actual count is very large, an off-by-1 error matters less than if the actual count is 1 or 2. **Asymmetric gradient:** the gradient is zero when $\hat{\lambda} = y$ but the gradient is different when $\hat{\lambda} = y - 1$ and $\hat{\lambda} = y + 1$, where the absolute gradient would be $1/(y-1)$ and $1/(y+1)$ respectively. The gradient tends to focus on penalizing the under-estimation rather than over-estimation.

In addition, we introduce a classification loss, $L_{cls}$, and a regularizer $L_{L1} = \sum_{j=1}^{N_B} \frac{1}{N_B} \left\| \overline{\mathbf{P}}_{ij} \right\|_{L1}$ to make our model more robust to sparse bags Tsai & Lin (2019) and ensure the inter-class separability of the learned representations Narayan et al. (2019). $L_{cls}$ is nothing but a binary cross-entropy loss to measure whether the network can classify if a class is present/absent in a given bag. We use a unified loss function defined as

$$
L_{LLC} = L_{reg} + L_{cls} + \beta L_{L1}
$$

(12)

### 5.1.2 ESTIMATING CLASS COUNT

Though we have $\hat{\lambda}_j$ as the expected count for class $j$, it remains a problem to estimate the exact count for each class. Mathematically, let the $\overline{\mathbf{c}}_i^j \in \mathbb{R}$ be the expected count for class $j$ in bag $b_i$, and the $t_i \in \Delta_K = \left\{ c \in \mathbb{Z}_+^K : \sum_{j=1}^K c_i^j = N_B \right\}$, then the exact count for each class in $b_i$ could be obtained by

$$
\hat{t}_i = \arg\max_{t \in \Delta_K} \sum_{j=1}^K \log \left( p \left( c_i^j = t_i \mid \overline{\mathbf{c}}_i^j \right) \right)
$$

(13)

This is constrained convex optimization and we can use any greedy algorithm to get the optimal solution. Initializing $\hat{t}_i$ as a vector of zeros, for each iteration, we manually calculate the marginal gain for increasing the $\hat{t}_i^j$ to $\hat{t}_i^j + 1$. We choose to increase the count of a class such that the increment is maximized, and iterate until the summation of the count vector is equal to $N_B$. By using a heap to track the maximum value, we can design an algorithm with time complexity of $O(log(K) * N_B)$ as is shown in Algorithm 1 in Appendix

### 5.2 STAGE-2: ESTIMATING INSTANCE LABEL (DECODER)

In Stage-1, the network outputs the expected count for each class, and then we develop an optimization problem to translate the expected count to exact count. In stage-2, given the exact count of each class, and the predicted probability distribution of each instance $p_\theta(y|b_{ij})$, we devise an assignment problem to get the instance-level label. Thus, we have the following linear-sum optimization problem.

$$
\max_x \sum_j^{N_B} \sum_k^K log(p_{ij}^k) * x_{ij}
$$
$$
s.t
$$
$$
\sum_k x_{ij}^k = 1, \forall j \in \{0, 1, \cdots N_B - 1\}
$$
$$
\sum_j x_{ij}^k = \overline{c_i}^k, \forall k \in \{0, 1, \cdots K - 1\}
$$
$$
p_{ij}^k \geq 0
$$
$$
x_{ij}^k \in \{0, 1\}, \forall j, k
$$

(14)

Table 1: Generated Data Summaries

| Distribution | Bag Size | lambda | # of training bags | # of testing bags | Avg. Count | Avg. (Std) Sparsity | Dataset Id |
|---|---|---|---|---|---|---|---|
| | 2 | 0.5 | 40,000 | 8,000 | 1.13 | 82%(4%) | p0 |
| | 4 | 0.5 | 22,000 | 4,500 | 1.28 | 68%(7%) | p1 |
| | 8 | 1.2 | 10,000 | 2,000 | 1.63 | 50.02%(10.53%) | p2 |
| | 16 | 2 | 4,000 | 1,000 | 2.28 | 28.9%(12.4%) | p3 |
| Poisson | 32 | 3.2 | 2,000 | 800 | 3.65 | 8.72%(8%) | p4 |
| | 8 | 1.2 | 5,000 | 1,000 | 3.43 | 8.32%(8%) | p5 |
| | 8 | 1.2 | 1,000 | 200 | 3.56 | 9%(8.2%) | p6 |
| | 16 | 8 | 4,000 | 1,000 | 6.52 | 79%(6%) | p7 |
| | 16 | 2 | 2,000 | 500 | 6.32 | 80%(6.2%) | p8 |
| | 16 | 2 | 1,000 | 200 | 6.21 | 79%(6.1) | p9 |
| Exponential | 8 | 0.67 | 10,000 | 2,000 | 1.97 | 58%(11.7%) | e0 |
| | 16 | 0.5 | 4,000 | 1,000 | 2.89 | 42%(13.9) | e1 |
| Uniform | 8 | N/A | 10,000 | 2,000 | 1.41 | 42%(9.21%) | u0 |
| | 16 | N/A | 4,000 | 1,000 | 1.95 | 18.7%(9.72%) | u1 |

where $p_{ij}^k = p_\theta(y = k|b_{ij})$ , $\bar{c}_i^k$ is the estimated exact count of class $k$ in bag $i$ and $x_{ij}^k = 1$ if and only if instance $j$ is assigned with label $k$.

# 6 EXPERIMENTS AND RESULTS

## 6.1 DATASETS

We conduct experiments based on the CIFAR-10 dataset. CIFAR-10 dataset consists of 50000 train and 10000 test images with each class consisting of 5000 and 1000 training and testing images respectively. We create bags of different sizes from each of the dataset images with and without replacement in our analysis. Each bag was created based on the size ranging from 2, 4, 6, 8, 16,32 bag size. In the case of the images with the replacement, we make sure that the number of images being repeated is restricted with a reuse parameter which is controllable to avoid a bag being over-represented by the same image. For each dataset generated with a different bag size, we ensure that the parameter configuration is set such that the maximum amount of the CIFAR-10 dataset both during the training and testing phase is used to represent the newly generated dataset with semi-weak labels. As CIFAR-10 dataset has a training to testing image ratio is maintained as 5:1, the same ratio is maintained in different datasets generated as shown in table 1

We generate the dataset with the following distribution: Poisson, exponential and uniform distribution. This distribution is based on how the class information is distributed in each given bag. The rationale with the generation of the instances of the bag in the above format was to ensure that the bags are generated based on naturally occurring instances in nature for the Poisson distribution, the exponential distribution, and the uniform distribution. The generation process is the following. 1) randomly sample a class $i$ with equal probability. 2) sample a number $n_i$ from the given distribution and then truncate it into the range between 0 and $N_B$. 3) sample $n_i$ instances from class $i$. In the case where the number of instances needed would be more than the total number of instances, the instances would be sampled at most two times. For each parameter setting, we sample the bags with 5 different random seeds. All the experiments are conducted on these 5 trials and the final value is obtained by averaging the best performance on the validation set. In the standard setting, to try to generate more balanced data, where each class has some instances in a bag. Therefore, we selective choose a hyperparameter for the distribution that minimizes the sparsity level of the bag. We define the sparsity of a bag as the number of absent class divided by the number of classes. Particularly, we use the following settings for Poisson distribution $[(N_B = 2, \lambda = 0.5), (N_B = 4, \lambda = 0.5), (N_B = 8, \lambda = 1.2), (N_B = 16, \lambda = 2.0), (N_B = 32, \lambda = 3.2)]$, and $[(N_B = 8, \lambda = 0.67), (N_B = 16, \lambda = 0.5)]$ The data summary table is provided in Table 1. For uniform distributed samples, the directly sample $N_B$ number of samples from the original dataset and label them with the counting vector.

## 6.2 ARCHITECTURE

We use a Residual Network with 18 layers as our base backbone for extracting features from the image. Given a bag of images $\{x_0, x_1, ..., x_{N_B}\} \in \mathcal{B_N}^{L \times H \times C}$, the embedding $e_i \in \mathcal{R}^{B_N}$ for each

instance is firstly extracted using a Convolutional Neural Network as the feature extractor. Then a linear layer $fc : \mathcal{R}^d \to \mathcal{R}^K$ is used to create a class activation map. Different pooling layer is applied at the end to generate counting prediction vector and logits for multi-label classification problem. All models are trained using Stochastic Gradient Descent with an initial learning rate of 0.01 for 100 epochs. The learning rate would be manually divided by 10 at epoch 30 and 50. Weight-decay is set to be 5e-4 for training. Standard data augmentations are utilized to avoid overfitting of CIFAR10 dataset, including random crop with padding of 4 and random horizontal flip with a probability of 0.5.

## 6.3 BASELINES

**Fully-supervised Upper-bound**: We want to argue that comparable results could be achieved by using semi-weak label compared to strong labels. Therefore, we also train ResNet18 in a fully supervised setting on CIFAR10. We trained it 250 epochs with initial learning rate of 0.1 and then divided it by 10 in the mid-training. This model could be deemed as an upper-bound of the performance in semi-weak setting. For our analysis with different base network architectures, we further use similar hyper-parameter

**Weakly Supervised Baseline**: We produce results with the weakly supervised baseline where we consider the loss from counting to be absent and thus we generate the results based on bag level prediction where only presence or absence of the class is available.

**Learning from Proportions**: If we set the loss function as KL-loss, the our model without the decoder could be used a baseline model to represent the performance of modeling the learning from counts as LLP.

## 6.4 EVALUATION

We use precision to evaluate our framework for both instance-level prediction and bag-level prediction. For bag-level prediction, we compute the precision rate using macro averaging. For fully supervised model, we predict the class label individually and label the bag positive for class $i$ if there is at lease one predicted instance for class $i$ in this bag. Similarly, for semi-weakly supervised model, we label a bag according to the instance-level predicted results.

Table 1 provides a detailed summary of all the generated dataset configurations. We use the following settings for the Bag size where $N_B \in [2, 4, 8, 16, 32]$ and the number of training instances are chosen such that we utilize most of the data available in the CIFAR-10 dataset. The average count here represents the aggregated mean of average number of instances per class. We represent the sparsity based on the percentage of classes with zero instance in a bag. Our experimental results for all the dataset configurations from Table 1 are summarized in Table 2. To remove the effect of randomness, for each dataset setting, we train the model with 5 different random seeds and then average the results across all trials. The comparison between baselines are discussed in the loss ablation section A.7

Table 2: Benchmarking Results on Different Datasets using Semi-weak label

| Dataset ID | Bag Prec. | Inc. Prec. | Dataset ID | Bag Prec. | Inc. Prec. |
|---|---|---|---|---|---|
| p0 | 94.24 | 92.76 | p7 | 64.37 | 85.20 |
| p1 | 93.78 | 92.65 | p8 | 89.92 | 81.92 |
| p2 | 93.20 | 91.21 | p9 | 84.97 | 69.93 |
| p3 | 92.92 | 88.07 | e0 | 90.69 | 91.05 |
| p4 | 93.9 | 71.91 | e1 | 87.10 | 87.65 |
| p5 | 90.54 | 87.73 | u0 | 94.86 | 91.42 |
| p6 | 77.39 | 68.91 | u1 | 95.62 | 87.34 |

## 6.5 BASELINE RESULTS

**Fully-supervised baseline**:Table 3 shows the results obtained with the different classifiers constructed as the base classifier for the classification with the CIFAR-10 dataset. We find that the bag

level prediction on the dataset p2 is comparable to the fully-supervised dataset which is expected as the semi-weak label dataset will be upper-bounded by the fully supervised setting.

Table 3: Fully supervised upper-bound results with classifier configuration

| Classifier | Loss | Bag Prec. | Inc. Prec. |
|---|---|---|---|
| ResNet18 | Poisson | 94.26 | 94.40 |
| ResNet34 | Poisson | 94.67 | 94.681 |
| ResNet50 | Poisson | 95.2 | 96.41 |
| MobileNetV2 | Poisson | 90.03 | 89.066 |

**Learning-from-proportion baseline**: For the learning-from-proportion baseline, results are shown in Table 7 (KL v.s Poisson) because we use KL-loss to proxy the baseline model of learning-from-proportion. This baseline achieved 87% instance-level precision, which is less than our proposed model on dataset p3, i.e., 88.07%. This result also holds if we consider bag size equals to 8 for dataset p2.

Table 4: Effect of the Choices of $L_{reg}$ on two standard dataset setting with $N_B \in \{8, 16\}$

| Dataset ID | $L_{reg}$ | Bag Prec. | Inc. Prec. |
|---|---|---|---|
| p5 | KL | 90.26 | 90.40 |
| p5 | L1 | 92.67 | 90.681 |
| p5 | Poisson | 93.2 | 91.21 |
| p3 | KL | 87.03 | 87.066 |
| p3 | L1 | 92.01 | 85.97 |
| p3 | Poisson | 92.92 | 88.08 |

**Learning-from-weak-label baseline**: For the learning-from-weak-label baseline, we report those number in Table 5. Once we deactivate the counting loss, then the model is converted into the traditional weak label setting. In Table 5, without counting loss, the performance is much worse than our proposed semi-weak labeling framework. It only achieves 87% instance-level prediction precision, which is 5% worse than our proposed framework.

Table 5: Ablation Study of Removing Regression Loss and Classification Loss. ($\beta \in \{0, 1.0\}$)

| Dataset ID | Bag Prec. | Inc. Prec. |
|---|---|---|
| p2 | 93.20 | 91.21 |
| p2 (w/o $L_{cls}$) | 93.23 | 91.17 |
| p2 (w/o $L_{reg}$) | 90.92 | 87.39 |
| p8 | 92.92 | 88.07 |
| p8 (w/o $L_{cls}$) | 62.13 | 83.92 |
| p8 (w/o $L_{reg}$) | 41.81 | 72.04 |

## 7 CONCLUSION

In this paper, we propose a novel machine learning problem, namely, learning from counts. We propose a two-stage framework to do instance-level prediction given only a counting vector for a bag is available. We generated dataset from CIFAR10 for experimentation. We achieve comparable results with the fully-supervised setting and much better results than the weakly supervised setting. Additionally, we introduced a L1 regularization term that make our model robust to sparse bags and achieve marginally prediction improvement on sparse bags. We believe semi-weak labels to provide better insights in real-world tasks where counting information can be easily obtained and plan to extend our work on other kinds of data.

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

# A APPENDIX

## A.1 EXAMPLE OF COUNT ASSIGNMENT OPTIMIZATION

Figure 3 gives an example with bag size of 4 and class size 3. If we negate the reward, then the objective becomes minimizing the "cost", which is referred to a classical linear-sum assignment problem in bipartite graphs Crouse (2016).

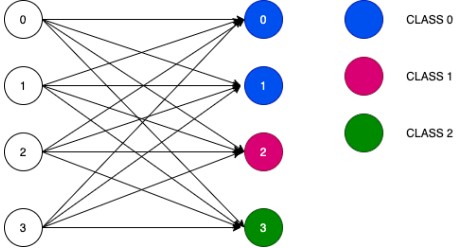

Figure 3: An example of assignment problem. The nodes on the left represent the unsigned instance and on the right represent the label. There are 4 instances in a bag and with 2 class-0 instances, 1 class-1 instance and 1 class-2 instance. Each edge has an associated reward $p_{ij}$, which is probability of instance $i$ belonging to class $j$. Each instance must have one assignment and the goal is to maximize the reward.

A.2  ALGORITHM

The following algorithm provides the greedy approach chosen to translate from the expected count to the exact count.

---
**Algorithm 1** Greedy Algorithm to Translate Expected Count to Exact Count
---
**Input**: Number of bag size $N_B$; Number of classes $K$; A list of floating points $[a_0, a_1, \cdots, a_K]$; a list of probability density function for Poisson distribution $[f_{a_0}(x), f_{a_1}(x), \cdots, f_{a_K}(x)]$
**Output**: A list of integers $[t_0, t_1, ..., t_K]$ such that $\sum_{j=0}^{K} t_j = N_B$.
**Initialize**: Let $q = heap <int, int>$ be a max-heap that stores pairs. Heap has two methods: $q.pop()$ would pop and return the max value from the heap and $q.push(v, i)$ would add a pair of $<v, i>$ to the heap. The comparison of pair is based the first value in the pair;
Let $ret = [0, 0, \cdots, 0]$ be a zero-initialized vector of size $K$;

 1: **for** $i = 1, 2, ....K$ **do**
 2:     $r \leftarrow f_{a_i}(ret[i] + 1) - f_{a_i}(ret[i])$
 3:     $q.push(r, i)$
 4: **end for**
 5: **for** $\_ = 2, ....N_B$ **do**
 6:     $v, i \leftarrow q.pop()$
 7:     $ret[i] \leftarrow ret[i] + 1$
 8:     $r \leftarrow f_{a_i}(ret[i] + 1) - f_{a_i}(ret[i])$
 9:     $q.push(r, i)$
10: **end for**
11: **return** $ret$

---

The following section provides the algorithm 2 for our semi-weak label classification system. In theory, the weak label classification system can work for all kinds of input data for classification whereas for the purpose of this paper we run experimental analysis on image classification task.

---
**Algorithm 2** Semi-Weak Label Classification System
---
**Input**: $batchsize$ number of bags $[(b_0, y_0^B), (b_1, y_1^B)..., (b_N, y_N^B)]$, loss weight $\alpha$, and regularization weight $\beta$, number of epochs $n_{epochs}$ and a loss function for regression $L_{reg}$
**Output**: A predicted label for each instance $[y_0, y_1, ..., y_N]$.

 1: **for** $n = 1, 2, ....n_{epochs}$ **do**
 2:     **for** $i = 1, 2, ....N_B$ **do**
 3:         Compute the class-wise distribution for $x_i$.
 4:     **end for**
 5:     **for** $k = 1, 2, ....K$ **do**
 6:         Compute $C_k = \sum_{i=1}^{N_B}$.
 7:     **end for**
 8:     Compute loss, $L_{reg}$ w.r.t label $y_B, C_k$.
 9:     Compute bag-level binary label using max pooling.
10:     Compute loss, $L_{cls}$ w.r.t label $y_B$.
11:     Compute the regularization term and compute the final loss given $\alpha, \beta$
12:     Update networks based on the combined loss.
13: **end for**
14: **for** $i = 1, 2, ....N$ **do**
15:     **for** $i = 1, 2, ....N_B$ **do**
16:         compute the the probability distribution for $x_i$ using learned classifier.
17:     **end for**
18:     Enumerate all combinations of counts and get the guess $t$ that maximizes the likelihood of $b_i$.
19:     Given $t$ and the probability distribution, using linear sum max algorithm to get the best assignment $y_0, y_1, ...y_{B_N}$
20: **end for**

---

A.3    RESULTS

A.4    ABLATION STUDY FOR THE DECODER

As is shown in Table 6, instead of using greedy search, we use the proposed three-stage framework to infer the instance-level labels. Results show consistent improvement after using the decoder algorithm. More importantly, when the bag is sparse, the counting for each class is high and thus the estimation would have high variance. Therefore, when the bag is less sparse, greedy search works well. But once the bag size increases or the bag becomes sparse, the high variance of the estimated logits would make the greedy solution less attractive. Therefore, the decoder contributes more to the performance of the predicted value ( $+2\%$ ) as the greedy predictions become unstable.

Table 6: Ablation Study for the Decoder

| Dataset ID | Inc. Prec. |
|---|---|
| p2 | 91.21 |
| p5 | 87.73 |
| p7 | 85.20 |
| p2(+Decoder) | 92.08 |
| p7(+Decoder) | 88.92 |
| p7(+Decoder) | 87.32 |

A.5    VARIATION WITH THE NUMBER OF TRAINING SAMPLES

We investigate the degree to which the size of the dataset relates to the model performance. We progressively sample different number of bags and evaluate them on the test set. We compare the results using different scales of training samples. The dataset settings are illustrated in Table 7.

Table 7: Scale Testing Results on two standard dataset setting with $N_B \in \{8, 16\}$.

| Dataset ID | Bag Prec. | Inc. Prec. |
|---|---|---|
| p2 | 93.2 | 91.21 |
| p5 | 90.54 | 87.74 |
| p6 | 77.39 | 68.91 |
| p3 | 92.93 | 88.08 |
| p8 | 89.92 | 81.92 |
| p9 | 84.97 | 69.93 |

A.6    VARIATION WITH THE BATCH SIZES AND ARCHITECTURES

We conduct experiments on variants batch sizes and find different performance characteristics. For all experiments, we train the model with batch size equal to 32, 64, 96, 128 and 256 as seen in Table 8 and bag size equal to 8 and 16 as the standard setting. We found that the results with the batch size doesn't provide large variation in the output of the optimization for the bag level training. However, as we increase the batch size there is a drop in performance observed on the instance level prediction.

A.7    EFFECTIVENESS OF DIFFERENT REGRESSION LOSSES

As is proposed for learning from proportions, KL loss is mostly widely used loss function for comparing the estimated class distribution and the true class distribution. Therefore, we also tried to use KL-loss as well L1 loss, which is usually used for multi instance regression as alternative choices.

As is summarized in Table 7, poisson loss function performs better than other loss functions for bag size equal to 8 and 16, in terms of bag level prediction as well as instance-level prediction. This is as expected as poisson distribution is naturally defined for non-negative and integer values like

Table 8: Ablation Study of Using different backbones and different batch sizes on dataset p2 $B_N = 8$.

| Backbone | Bag Prec. | Inc. Prec. |
|---|---|---|
| Resnet18 (bs=32) | 94.036 | 92.096 |
| Resnet18 (bs=64) | 93.20 | 91.21 |
| Resnet18 (bs=96) | 93.630 | 91.944 |
| Resnet18 (bs=128) | 93.2 | 91.23 |
| Resnet18 (bs=192) | 92.661 | 90.649 |
| Resnet18 (bs=256) | 92.541 | 90.183 |
| Resnet34 (bs=128) | 93.763 | 91.763 |
| Resnet50 (bs=128) | 93.512 | 91.893 |
| MobileNetV2 (bs=128) | 91.474 | 89.002 |

counting. Another explanation is that the data is generated from poisson distribution but according to Table 9, even though the dataset is generated from non-poisson distribution, the poisson loss function still achieve comparable values. This implies that poisson distribution is robust.

## A.8    EFFECT OF REGULARIZER WITH SPARSE BAGS

In order to analyze the model performance on bags with different sparsity, we further create bags with higher expected counts for each class to generate bags with higher sparsity. The specific setting and data statistics are summarized in Table 1

Interestingly, we found that the regularized term works really well for sparse bags. It consistently improve the performance thought marginally.

Table 9: Effect of the Regularizer on Sparse Bags (Dataset ID=p7)

| $\beta$ | Bag Prec. | Inc. Prec. |
|---|---|---|
| 0.0 | 63.7 | 84.67 |
| 0.01 | 64.37 | 85.20 |
| 0.1 | 65.09 | 85.52 |
| 0.5 | 65.07 | 85.87 |

## A.9    PERFORMANCES OVER DIFFERENT BAG SIZES

As we see in table 1, we have different bag sizes generated with a variation of the parameters lambda and number of training bags to create new copies of dataset with different bag size. Table 2 shows that as we increase the bag size the performance remains fairly increasing upto a bag size of 8 and then there is a gradual decrease in performance observed with large drop seen with the bag size of 32. Such variation can be attributed to the fact that with the larger bag size the nature of the distribution of the bags to the instance level information per class is artificial and there is a consistent decrease in the performance as we increase bag size from 16 to 32 and beyond.

## A.10    COMPARISON WITH DIFFERENT DISTRIBUTION

Besides generating data from poisson distribution, we also train our models with the exponential distribution as well as the uniform distribution. We observe that the results with the poisson distribution is better as compared the exponential distribution which is in accordance with our expectation as the count information in case of the exponential distribution will be sparse than case of the poisson distribution. The analysis can be reviewed in the table 2 for case of the dataset id e0 and p2 there is a stark difference with the same bag size and sparsity distribution but difference in the distribution of classes in each bag results in performance drop from 93.2% precision to 90.69% precision.

