# OpenReview forum: "Training image classifiers using Semi-Weak Label Data"
_ICLR.cc/2023/Conference — Submitted to ICLR 2023_

### Official Review · Reviewer_CGT7 · 2022-10-21

**Confidence:** 4
**Correctness:** 3
**Technical Novelty And Significance:** 2
**Empirical Novelty And Significance:** 1
**Recommendation:** 5

**Clarity, Quality, Novelty And Reproducibility:**

In general the paper is written well and the proposed method has some novelty. However, the novelty is limited in my opinion given that the authors built their method based on an existing method and just change the utilized distribution (using Poisson instead of exponential one). The experimental evaluation is also very weak, which degrades the contribution. The experiments can be reproduced if the authors share their codes.

**Strength And Weaknesses:**

The main strengths of the paper can be summarized as follows:
i) In general the paper is written well and the classification setting and the motivation are described in details.
ii) The proposed method seems novel.
iii) Good accuracies are reported on a simple dataset.
The main weaknesses of the paper can be summarized as follows:
i) Some of the used terms and statements should be corrected in the manuscript. The authors consider as a weak label when a label is given to an image yet they assume a label is strong if the bounding box information is given.  This is not correct. If the main goal is just image classification rather than visual object detection, a label associated to an image is enough and that is all we need. Therefore, it is also strong supervision. This kind of labeling can be considered as weak if the goal is visual object detection rather than classification since bounding box information is also necessary for this problem. These issues must be corrected.
ii) The authors use Poisson distribution in order to estimate the number of samples in classes. As written in the text, Poisson distribution is used to estimate the number of events occurring in a given time period, and basically 3 important conditions must be satisfied to use this distribution. One of the is the count of each class must be independent and the authors assume that is satisfied. What about other conditions and I wonder why this distribution is the best choice for his problem. Why not to use exponential distribution as in other methods (Quadrianto etal, 2009; Patrini et al. 2014)? More arguments must be given to support this claim.
iii) In the experimental setting, the authors use both discrete random variable distribution (passion) as well as continuous random variable distributions (exponential and uniform distributions) to generate the dataset splits. But, they argued that continuous distributions cannot be used for this goal. Please comment on this.
iv) The major limitation of the paper is the weakness of the experiments. The authors conduct tests only on a simple dataset Cifar-10. The images in this dataset have only one label which is far from real life. In fact, real-world images have multiple labels (i.e., many object may exist in images, for example there may be people, cars, buildings etc. in a given image). Therefore, instead of this simple datasets, it is better to conduct tests on more realistic datasets such as COCO, NUS-WIDE etc. In that case, more reliable results can obtained.
v) As a last comment, the improvement over the methods using label proportions is very minor, only 1.07%. This may not have any significancy and the performance difference may due to the chance factors.


**Summary Of The Paper:**

In this paper, the authors propose a classification method that uses semi-weak labels. In this setting, the authors assume that the number of instances in each class is known in addition to the presence or absence of a given class. To utilize this information, the authors propose a loss function utilizing Poisson distribution. The method is built based on an existing method called DLLP. The authors test their method on Cifar-10 dataset and report comparable results to the method that uses full supervised label information.

**Summary Of The Review:**

In general, the paper is written well. However, motivation for using Poisson distribution must be supported with theoretical arguments. In addition, the authors must conduct tests on more realistic and challenging datasets such MS-COCO.

---

### Official Review · Reviewer_b2Pz · 2022-10-23

**Confidence:** 4
**Correctness:** 3
**Technical Novelty And Significance:** 2
**Empirical Novelty And Significance:** 2
**Recommendation:** 3

**Clarity, Quality, Novelty And Reproducibility:**

- The paper is not very well-organized and confusing.
- The overall quality and novelty is limited due to the lack of motivation for the problem setting and the presentation of the proposed method.
- The repoducibility is not guaranteed due to the ambigous description of the algorithm.

**Strength And Weaknesses:**

Strength
- This paper draws attention on considering new problem settings for multiclass classification.

Weakness
- I could not understand the motivation of the problem setting. The authors give an example of annotating multiple dogs in an image. However, considerting image classification, an image containg multiple dogs is the instance level data, then what are bags in this case? Concrete examples should be give to motivate the problem setting.
- Knowing exact numbers of instances for each class seems like a very strong annotation. Are there actual cases that users know exact numbers but still not knowing labels for each instance?
- Sections 3.1, proposing a loss function, seems to be off the story and not related to the later part of the paper.
- The part describing the two-stage algorithm is hard to follow. I could not see a clear story to know what to do next.

**Summary Of The Paper:**

This paper considers a special case of multiclass classification problem. Here, bag level labels and counts of each class are available. The paper proposes a two-stage algorithm to solve the new defined problem.

**Summary Of The Review:**

I recommend to reject as there are many aspects of the paper can be improved, such as the motivation, the presentation of the proposed algorithm and the organization of the paper.

---

### Official Review · Reviewer_vbs2 · 2022-10-24

**Confidence:** 4
**Correctness:** 2
**Technical Novelty And Significance:** 2
**Empirical Novelty And Significance:** 2
**Recommendation:** 3

**Clarity, Quality, Novelty And Reproducibility:**

- This paper lacks clarity especially regarding mathematical presentation.
It includes not only inconsistent/confusing notations but also unclear formulation.
The authors should, if possible, follow the notations used in the related works of label proportion which address almost the same tasks.

- Novelty is limited due to the simple application of Poisson loss.

- The unclear experimental setting makes it hard to reproduce the results.


**Strength And Weaknesses:**

## Strength
+ Weak annotation of label counts is effectively exploited in the Poisson model to build a loss function.
+ The authors construct a weakly annotated dataset by assigning class counts to a bag of image samples drawn from Cifar-10.

## Weakness
### Limited technical novelty.
- Simply introducing a well-known Poisson loss in Eq.(10) lacks novelty, compared to the prior works of label proportion [Dulac-Arnold+19, Ardehaly+17].
In formulating the loss, it is hard to understand the sentence of "since we assume the size of the bag is unknown, we assume the count for each class in a bag is independent".
In this weakly-supervised setting using label counts, the size of bag is accessible and in particular is constant in the experiments of Sec.5.
Thus, there lacks an analysis or discussion to explain why such an independence assumption is valid.
As to the probabilistic formulation for label counts/proportions, an efficient approach is also proposed for small-sized bags in [R1].

[R1] Denis Barucic and Jan Kybic. FAST LEARNING FROM LABEL PROPORTIONS WITH SMALL BAGS. ICIP2022.

- The gradient in Eq.(11) would be unstable for \hat{\lambda} ~ 0. How do the authors cope with the issue? It would be necessary to analyze the stability of training based on the loss.

- L1-regularization for the posterior f(x_i) in Eq.(6) is unclear since it seems to make no sense in the loss. The posterior p(y|x) is by definition subject to a unit-sum constraint of \sum_y p(y|x), and thus its L1-norm always results in 1, which does not regularize the training at all.

- It is difficult to recognize the efficacy of the post-processing to finally predict bag-level/instance-level label in Secs.5.1.2&5.2.
How well does it actually work in comparison to the simplest approach of argmax over the instance logits?

- Related to the post-processing, the second constraint in Eq.(14), \sum_j x_{ij}^k = \bar{c}_i^k, is unclear.
\bar{c}_i^k is defined as real-value prediction while \sum_j x_{ij}\k is a binary-value (one-hot) variable to indicate a class label, thereby both of which are not mathematically compatible in the equality.

### Poor presentation.
* Mathematical forms are poorly presented in this paper.
For example, $\bigoplus$ usually stands for "direct-sum" but is used as just a summation, which is quite confusing; the authors should use a standard $\sum$, following a mathematical convention.
Besides, mathematical notations are employed in an inconsistent and confusing way across Secs.3~5.  Such a messy mathematical presentation makes it quite hard to follow the technical content.

* The experiments are organized in an unclear way.
While methods are evaluated on image classification tasks, the authors rarely mention the task in Secs.1~5 except for the "title"; it is even apart from the task in Fig.1 that the authors enthusiastically discussed. So, the motivation for the image classification tasks is unclear.
For constructing benchmark datasets, bags are sampled in a confusing fashion. In the previous works such as [Dulac-Arnold+19], the same type of dataset is built on the basis of Cifar-10 and so the authors should follow the experimental protocol, which also makes it possible to fairly compare the method with the previous approach.
Otherwise, one could have a doubt that such a confusing data construction based on Poisson prior is designed for bringing an unfair bias toward the proposed method.

* In experimental evaluation, the comparison methods are unclearly described. For example, in Table 3, what does the loss of "Poisson" mean? To train a fully supervised model using instance-level annotation, it is a common way to apply the Softmax cross-entropy loss in this Cifar-10 task.

* Unclear effectiveness of the proposed counting-based weak supervision.
This paper lacks sufficiently convincing stuffs to show the efficacy of the weakly-supervised classification tasks based on label counts that the authors present in this paper.
The story about acoustic event detection depicted by Fig. 1 and its related discussion seems to be convincing.
But, the technical part regarding the proposed method in Secs.3~5 is far from the detection task since the method is built upon the assumption that "sum of counts per category in each bag must equal the bag size" (page 2).
The acoustic detection task goes beyond the assumption and thus the method is inapplicable to the task.
Practical validity and feasibility of the assumption is not fully demonstrated in this paper; practically speaking, when we count class labels in a bag, we would have to identify a class category of each instance, being equivalent to demanding instance-level supervision.
Such an inconsistent discussion makes this paper difficult to follow.
Besides, without demonstrating the proposed weakly-supervised task in a feasible and convincing way, it is also difficult to find the meaningful benefit to address the task.

**Summary Of The Paper:**

This paper presents a method to train a classification model in a new weakly supervised framework. Similarly to the scenario of label proportions, the authors impose the assumption that a bag of instance samples is equipped only with class-frequency counts without annotating each instance. The information of label counts is then leveraged to formulate an objective loss function by means of Poisson loss. In the experiments, the authors build a benchmark dataset based on Cifar-10 by following the weakly-supervised setting, and empirically evaluate the proposed method.

**Summary Of The Review:**

Due to lack of clarity and technical novelty, this paper leans toward rejection.

---

### Decision · Program_Chairs · 2023-01-20

**Decision:**

Reject

**Justification For Why Not Higher Score:**

All the reviewers give reject ratings. They have concerns on the motivation of the proposed setting, the novelty of the proposed method, the  presentation quality. Reviewers also ask for additional technical details and clarifications. However, the authors do not provide any further response to these questions.

**Justification For Why Not Lower Score:**

N/A

**Metareview: Summary, Strengths And Weaknesses:**

This paper introduces a new problem setting for multi-class classification, where the label counts are provided for a bag of training samples. Then a Poisson based loss function is designed for training the classification model under this setting. They conducted experiments on a customized CIFAR dataset to evaluate effectiveness of the proposed method.

Strength:

- the paper aims to explore a new setting for multi-class classification problems.

Weakness:

- the proposed new setting is not well motivated. It is hard to justify whether it is a valid setting in the realistic applications.
- the proposed solution has limited novelty. The similar loss function design has been proposed by existing works.
- the experiments are not sufficient.